# DIFFUSION-BASED REPRESENTATION LEARNING

## ABSTRACT

Diffusion-based methods represented as stochastic differential equations on a continuous time domain have recently proven successful as a non-adversarial generative model. Training such models relies on denoising score matching, which can be seen as multi-scale denoising autoencoders. Here, we augment the denoising score matching framework to enable representation learning without any supervised signal. GANs and VAEs learn representations by directly transforming latent codes to data samples. In contrast, the introduced diffusion-based representation learning relies on a new formulation of the denoising score matching objective and thus encodes information needed for denoising. We illustrate how this difference allows for manual control of the level of details encoded in the representation. Using the same approach, we propose to learn an infinite-dimensional latent code which achieves improvements of state-of-the-art models on semi-supervised image classification. As a side contribution, we show how adversarial training in diffusion-based models can improve sample quality and improve sampling speed using a new approximation of the prior at smaller noise scales.

## 1 INTRODUCTION

Diffusion-based models have recently proven successful for generating images (Sohl-Dickstein et al. (2015); Song & Ermon (2020); Song et al. (2020)), graphs (Niu et al. (2020)), shapes (Cai et al. (2020)), and audio (Chen et al. (2020b); Kong et al. (2021)). Two promising approaches apply step-wise perturbations to samples of the data distribution until the perturbed distribution matches a known prior (Song & Ermon (2019); Ho et al. (2020)). A model is trained to estimate the reverse process, which transforms samples of the prior to samples of the data distribution (Saremi et al. (2018)). Diffusion models were further refined (Nichol & Dhariwal (2021); Luhman & Luhman (2021)) and even achieved better image sample quality than GANs (Dhariwal & Nichol (2021); Ho et al. (2021); Mehrjou et al. (2017)). Further, Song et al. showed that these frameworks are discrete versions of continuous-time perturbations by stochastic differential equations and proposed a diffusion-based generative modeling framework on continuous time. Unlike generative models such as GANs and various forms of autoencoders, the original form of diffusion models does not come with a fixed architectural module that captures the representation.

Learning desirable representations has been an integral component of generative models such as GANs and VAEs (Bengio et al. (2013); Radford et al. (2016); Chen et al. (2016); van den Oord et al. (2017); Donahue & Simonyan (2019); Chen et al. (2020a); Schölkopf et al. (2021)). Considering diffusion-based methods as promising and theoretically grounded generative models, here we propose a method to augment the underlying Stochastic Differential Equation (SDE) for learning a latent data-generating code. The key idea is to provide a representation of the clean data as additional input to the model estimating the score function. Our approach is illustrated in Figure 1. Recent works on visual representation learning achieve impressive performance on the downstream task of classification by applying contrastive learning (Chen et al. (2020c); Grill et al. (2020); Chen & He (2020); Caron et al. (2021)). However, training the encoder to output similar representation for different views of the same image removes information about the applied augmentations, thus the performance benefits are limited to downstream tasks that do not depend on the augmentation, which has to be known beforehand. For this reason, we do not apply contrastive learning in order not to restrict the learned representation to specific downstream tasks and solve a more general problem instead. We provide a summary of contrastive learning approaches in A.1. Similar to our approach, Denoising Autoencoders (DAE) (Vincent et al. (2008)) can be used to encode representations that

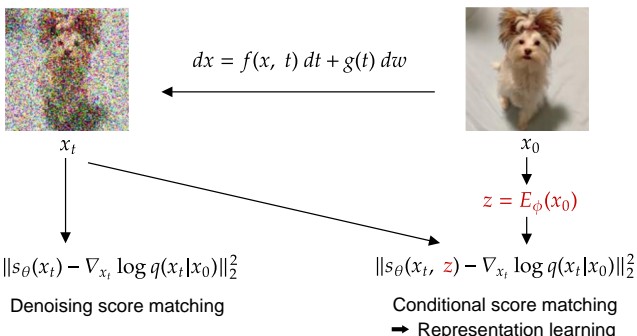

Figure 1: Conditional score matching with a parametrized latent code is representation learning. Denoising score matching estimates the score at each $x_t$; we add a latent representation $z$ of the clean data $x_0$ as additional input to the score estimator.

can be manually controlled by adjusting the noise scale (Geras & Sutton (2015); Chandra & Sharma (2014); Zhang & Zhang (2018)). Note that unlike DAEs, the encoder in our approach does not receive noisy data as input, but instead extracts features based on the clean images. For example, this key difference allows DRL to be used to limit the encoding to fine-grained features when focusing on low noise levels, which is not possible with DAEs. Even though diffusion models are considered non-adversarial, in this work we show that it can actually be an adversarial process by means of the weighting function $\lambda$ which has positive effect in the sampling quality of the diffusion-based models.

We begin by briefly revisiting the foundations of diffusion-based generative models in Section 1.1. In Section 2 we present our method for representation learning, propose how to apply adversarial training in diffusion-based models in Section 3.1, and motivate the use of fewer sampling steps in Section 3.2. We follow up with experimental results and evaluations of our proposed methods in Section 4. For further clarity, the main contributions of this work are itemized in the following.

- We present an alternative formulation of the denoising score matching objective, showing that the objective cannot be reduced to zero.
- We introduce and evaluate Diffusion-based Representation Learning (DRL), a new framework for representation learning in diffusion-based generative models. We show how this framework allows for manual control of the level of details encoded in the representation. We extend our approach to an infinite-dimensional code and evaluate it on the downstream task of semi-supervised image classification, improving state-of-the-art approaches.
- Unlike the widely admitted non-adversarial nature of these models, we show there exists an inherent component in the formulation that acts adversarially and can be leveraged to improve the sample quality.
- We evaluate the effect of the initial noise scale and achieve significant improvements in sampling speed, which is a bottleneck in diffusion-based generative models compared with GANs and VAEs, without sacrificing image quality.

## 1.1 DIFFUSION-BASED GENERATIVE MODELING

In the following, we give a brief overview of the technical background for the framework of the diffusion-based generative model, for example, as described in (Song et al., 2021b). The forward diffusion process of the data is modeled as a SDE on a continuous time domain $t \in [0, T]$. Let $x_0 \in \mathbb{R}^d$ denote a sample of the data distribution $x_0 \sim p_0$, where $d$ is the data dimension. The trajectory $(x_t)_{t \in [0,T]}$ of data samples is a function of time determined by the diffusion process. The SDE is chosen such that the distribution $p_{0T}(x_T|x_0)$ for any sample $x_0 \sim p_0$ can be approximated by a known prior distribution. Notice that the subscript $0T$ of $p_{0T}$ refers to the conditional distribution of the diffused data at time $T$ given the data at time 0. For simplicity we limit the remainder of this paper to the so-called Variance Exploding SDE (Song et al. (2021b)), that is,

$$\mathrm{d}x = f(x, t)\,\mathrm{d}t + g(t)\,\mathrm{d}w := \sqrt{\frac{\mathrm{d}[\sigma^2(t)]}{\mathrm{d}t}}\,\mathrm{d}w, \qquad (1)$$

where w is the standard Wiener process. The perturbation kernel of this diffusion process has a closed-form solution being $p_{0t}(x_t|x_0) = \mathcal{N}(x_t; x_0, [\sigma^2(t) - \sigma^2(0)]I)$. It was shown by Anderson (1982) that the reverse diffusion process is the solution to the following SDE:

$$\mathrm{d}x = [f(x, t) - g^2(t)\nabla_x \log p_t(x)] \mathrm{d}t + g(t) \mathrm{d}\overline{w}, \tag{2}$$

where $\overline{w}$ is the standard Wiener process when the time moves backwards. Thus, given the score function $\nabla_x \log p_t(x)$ for all $t \in [0, T]$, we can generate samples from the data distribution $p_0(x)$. In order to learn the score function, the simplest objective is Explicit Score Matching (ESM) (Hyvärinen & Dayan (2005)), that is,

$$\mathbf{E}_{x_t}\left[\|s_\theta(x_t, t) - \nabla_{x_t} \log p_t(x_t)\|_2^2\right]. \tag{3}$$

Since the ground-truth score function $\nabla_{x_t} \log p_t(x_t)$ is generally not known, one can apply denoising score matching (DSM) (Vincent (2011)), which is defined as the following:

$$J_t^{DSM}(\theta) = \mathbf{E}_{x_0}\{\mathbf{E}_{x_t|x_0}[\|s_\theta(x_t, t) - \nabla_{x_t} \log p_{0t}(x_t|x_0)\|_2^2]\}. \tag{4}$$

The training objective over all $t$ is augmented by Song et al. (2021b) with a time-dependent positive weighting function $\lambda(t)$, that is, $J^{DSM}(\theta) = \mathbf{E}_t\left[\lambda(t)J_t^{DSM}(\theta)\right]$.

## 2 DIFFUSION-BASED REPRESENTATION LEARNING

### 2.1 ALTERNATIVE FORMULATION OF DENOISING SCORE MATCHING

We begin this section by presenting an alternative formulation of the Denoising Score Matching (DSM) objective, which shows that this objective cannot be made arbitrarily small. Formally, the formula of the DSM objective can be rearranged as

$$J_t^{DSM}(\theta) = \mathbf{E}_{x_0}\{\mathbf{E}_{x_t|x_0}\left[\|\nabla_{x_t} \log p_{0t}(x_t|x_0) - \nabla_{x_t} \log p_t(x_t)\|_2^2\right.$$
$$\left. + \|s_\theta(x_t, t) - \nabla_{x_t} \log p_t(x_t)\|_2^2\right]\}. \tag{5}$$

The above formulation holds, because the DSM objective in equation 4 is minimized when $\forall x_t : s_\theta(x_t, t) = \nabla_{x_t} \log p_t(x_t)$, and differs from ESM in equation 3 only by a constant (Vincent (2011)). Hence, the constant is equal to the minimum achievable value of the DSM objective. A detailed proof is included in the Appendix (A.2). It is noteworthy that the first term in the rhs of the equation 5 does not depend on the learned score function of $x_t$ for every $t \in [0, T]$. Rather, it is influenced by the diffusion process that generates $x_t$ from $x_0$. This observation has not been emphasized previously, probably because it has no direct effect on the learning of the score function that is handeled by the second term in the rhs of equation 5. However, the additional constant has major implications for finding other hyperparameters such as the function $\lambda(t)$ and the choice of $\sigma(t)$ in the forward SDE. To the best of our knowledge, there is no known theoretical justification for the values of $\sigma(t)$. While these hyperparameters could be optimized in ESM using gradient-based learning, this ability is severely limited by the non-vanishing constant in equation 5. Similarly, it impacts the behaviour of adversarial training in diffusion models, which we analyze in Section 4.2.

Even though the non-vanishing constant in the denoising score matching objective presents a burden in multiple ways such as hyperparameter search and model evaluation, it provides an opportunity for latent representation learning, which will be described in the following sections.

### 2.2 CONDITIONAL SCORE MATCHING

Class-conditional generation can be achieved in diffusion-based models by training an additional time-dependent classifier $p_t(y|x_t)$ (Song et al. (2021b)). In particular, the conditional score for a fixed $y$ can be expressed as the sum of the unconditional score and the score of the classifier, that is, $\nabla_{x_t} \log p_t(x_t|y) = \nabla_{x_t} \log p_t(x_t) + \nabla_{x_t} \log p_t(y|x_t)$. We propose conditional score matching as an alternative way to allow for controllable generation. Given supervised samples $(x, y(x))$, the new training objective for each time $t$ becomes

$$J_t^{CSM}(\theta) = \mathbf{E}_{x_0}\{\mathbf{E}_{x_t|x_0}[\|s_\theta(x_t, t, y(x_0)) - \nabla_{x_t} \log p_{0t}(x_t|x_0)\|_2^2]\}. \tag{6}$$

The objective in equation 6 is minimized if and only if the model equals the conditional score function $\nabla_{x_t} \log p_t(x_t|y(x_0) = \hat{y})$ for all labels $\hat{y}$.

### 2.3 LEARNING LATENT REPRESENTATIONS

Since supervised data is limited and rarely available, we propose to learn a labeling function $y(x_0)$ at the same time as optimizing the conditional score matching objective in equation 6. In particular, we represent the labeling function as a trainable encoder $E_\phi : \mathbb{R}^d \to \mathbb{R}^c$, where $E_\phi(x_0)$ maps the data sample $x_0$ to its corresponding code in the $c$-dimensional latent space. The code is then used as additional input to the model. Formally, the proposed learning objective for Diffusion-based Representation Learning (DRL) is the following:

$$J^{DRL}(\theta, \phi) = \mathbf{E}_{t,x_0,x_t}[\lambda(t)\|s_\theta(x_t, t, E_\phi(x_0)) - \nabla_{x_t} \log p_{0t}(x_t|x_0)\|_2^2]. \tag{7}$$

To get a better idea of the above objective, we provide an intuition for the role of $E_\phi(x_0)$ in the input of the model. The model $s_\theta(\cdot, \cdot, \cdot) : \mathbb{R}^d \times \mathbb{R} \times \mathbb{R}^c \to \mathbb{R}^d$ is a vector-valued function whose output points to different directions based on the value of its third argument. In fact, $E_\phi(x_0)$ selects the direction that best recovers $x_0$ from $x_t$. Hence, when optimizing over $\phi$, the encoder learns to extract the information from $x_0$ in a reduced-dimensional space that helps recover $x_0$ by denoising $x_t$. Notice that finding the denoising direction requires information from both $x_0$ and $x_t$ and $E_\phi$ can only extract the partial information from the source $x_0$.

We show in the following that equation 7 is a valid representation learning objective. The score of the perturbation kernel $\nabla_{x_t} \log p_{0t}(x_t|x_0)$ is a function of only $t$, $x_t$ and $x_0$. Thus the objective can be reduced to zero if all information about $x_0$ is contained in the latent representation $E_\phi(x_0)$. When $E_\phi(x_0)$ has no mutual information with $x_0$, the objective can only be reduced up to the constant in equation 5. Hence, our proposed formulation takes advantage of the non-zero lower-bound of equation 5, which can only vanish when data information is distilled in a code provided as input to the model. These properties show that equation 7 is a valid objective for representation learning.

Our proposed representation learning objective enjoys the continuous nature of SDEs, a property that is not available in many previous representation learning methods (Radford et al. (2016); Chen et al. (2016); Locatello et al. (2019)). In DRL, the encoder is trained to represent the information needed to denoise $x_0$ for different levels of noise $\sigma(t)$. We hypothesize that by adjusting the weighting function $\lambda(t)$, we can manually control the granularity of the features encoded in the representation and provide empirical evidence as support. When $t \to T$ that is associated to high noise level, the mutual information of $x_t$ and $x_0$ starts to vanish, thus denoising requires all information about $x_0$ to be contained in the code. In contrast, when $t \to 0$ that corresponds to low noise levels, $x_t$ contains coarse-grained features of $x_0$ and only fine-grained properties may have been washed out due to the small magnitude of noise. Hence, the representation learns to keep the information needed to recover these fine-grained details. We provide empirical evidence to support this hypothesis in Section 4.

It is noteworthy that $E_\phi$ does not need to be a deterministic function. In principle, it can be viewed as an information channel that controls the amount of information that the diffusion model receives from the initial point of the diffusion process. With this perspective, any deterministic or stochastic function that can manipulate $I(x_t, x_0)$, the mutual information between $x_0$ and $x_t$, can be used. This opens up the room for stochastic encoders similar to VAEs that we call Variational Diffusion-based Representation Learning (VDLR) from this point onward. The formal objective of VDLR is

$$J^{VDRL}(\theta, \phi) = \mathbf{E}_{t,x_0,x_t}[\mathbf{E}_{z \sim E_\phi(Z|x_0)}[\lambda(t)\|s_\theta(x_t, t, z) - \nabla_{x_t} \log p_{0t}(x_t|x_0)\|_2^2] \tag{8}$$

$$+ \mathcal{D}_{KL}(E_\phi(Z|x_0)\|\mathcal{N}(Z; 0, I)). \tag{9}$$

### 2.4 INFINITE-DIMENSIONAL REPRESENTATION OF FINITE-DIMENSIONAL DATA

We now present an alternative version of DRL where the representation is a function of time. Instead of emphasizing on different noise levels by weighting the training objective, as done in the previous section, we can provide the time $t$ as input to the encoder. Formally, the new objective is the following:

$$\mathbf{E}_{t,x_0,x_t}[\lambda(t)\|s_\theta(x_t, t, E_\phi(x_0, t)) - \nabla_{x_t} \log p_{0t}(x_t|x_0)\|_2^2], \tag{10}$$

where $E_\phi(x_0)$ in equation 7 is replaced by $E_\phi(x_0, t)$. Intuitively, it allows the encoder to extract the necessary information of $x_0$ required to denoise $x_t$ for any noise level. This can be seen as a rich representation learning in the following way. Normally in autoencoders or other *static* representation

learning methods, the input data $x_0 \in \mathbb{R}^d$ is mapped to a single point $z \in \mathbb{R}^c$ in the code space. We propose a richer representation where the input $x_0$ is mapped to a curve in $\mathbb{R}^c$ instead of a single point. Hence, the learned code is produced by the map $x_0 \to (E_\phi(x_0, t))_{t \in [0,T]}$ where the infinite-dimensional object $(E_\phi(x_0, t))_{t \in [0,T]}$ is the encoding for $x_0$.

**Proposition 1.** *For any downstream task, the infinite-dimensional code $(E_\phi(x_0, t))_{t \in [0,T]}$ learned using the objective in equation 10 is at least as good as finite-dimensional static codes learned by the reconstruction of $x_0$.*

*Proof sketch.* The score matching objective can be seen as a reconstruction objective of $x_0$ conditioned on $x_t$. The terminal time $T$ is chosen large enough so that $x_T$ is independent of $x_0$, hence the objective for $t = T$ is equal to a reconstruction objective without conditioning. Therefore, there exists a $t \in [0, T]$ where the learned representation $E_\phi(x_0, t)$ is the same representation learned by the reconstruction objective of a vanilla autoencoder. □

The full proof for Proposition 1 can be found in the Appendix (1). A downstream task can leverage this rich encoding in various ways, including the use of either the static code for a fixed $t$, or the use of the whole trajectory $(E_\phi(x_0, t))_{t \in [0,T]}$ as input. We posit the conjecture that the proposed rich representation is helpful for downstream tasks when used for pretraining, where the value of $t$ is either a model selection parameter or jointly optimized with other parameters during training, and evaluate it on semi-supervised image classification in Section 4.1.1. The current state-of-the-art model for many semi-supervised image classification benchmarks is LaplaceNet (Sellars et al. (2021)). It alternates between assigning pseudo-labels to samples and supervised training of a classifier. The key idea is to assign pseudo-labels by minimizing the graphical Laplacian of the prediction matrix, where similarities of data samples are calculated on a hidden layer representation in the classifier. Note that LaplaceNet applies *mixup* (Zhang et al. (2017)) that changes the input distribution of the classifier. We evaluate our method both with and without mixup on CIFAR-10 (Krizhevsky et al. (a)), CIFAR-100 (Krizhevsky et al. (b)) and MiniImageNet (Vinyals et al. (2016)).

## 3 REMARKS ON DIFFUSION-BASED GENERATIVE MODELS

The following sections contain side-contributions on training and sampling in diffusion-based models. First we want to note that diffusion models are claimed to minimize KL-Divergence when using the respective likelihood weighting (Song et al. (2021a)). However, the assumption of the model being curl-free is usually not enforced and thus this statement might not hold in practice.

### 3.1 ADVERSARIAL TRAINING IN DIFFUSION-BASED GENERATIVE MODELS

In general, diffusion-based generative models enjoy the advantages of non-adversarial training. Hence, they do not suffer from mode collapse, a common problem observed in GANs (Thanh-Tung et al. (2018)), but instead can be trained to minimize KL-Divergence. However, it has been shown that KL-Divergence is not a good indicator of perceptual image quality (Theis et al. (2016); Gatys et al. (2017); Arjovsky et al. (2017)). Hence, GANs were extended to maximize the variation lower bound of the family of $f$-divergences (Nowozin et al. (2016)). Similarly, Song et al. introduced the set of $\lambda$-divergences between two probability distributions $p$ and $q$, defined as

$$D_\lambda(p||q) = \frac{1}{2} \int_0^T \mathbf{E}_{x \sim p_t(x)}[\lambda(x, t)\|\nabla_x \log p_t(x) - \nabla_x \log q_t(x)\|_2^2]dt. \tag{11}$$

Note that $\lambda$-divergences can express any $f$-divergence and that we can train on the respective divergence solely by adjusting $\lambda(x, t)$ in the denoising score matching objective. However, minimizing a specific $f$-divergence requires the knowledge of the ratio of densities $p_t(x)/q_t(x)$ that is not at hand.

Previous work on adversarial training in diffusion models added a discriminator loss training the model to output visually appealing samples after each denoising step (Jolicoeur-Martineau et al. (2020)). In contrast, we form a min-max game to minimize the worst-case $\lambda$-divergence defined by an additional adversary, which is trained alternately with the model. Since we do not have access to the score function $\nabla_{x_t} \log p_t(x_t)$, we approximate the $\lambda$-divergence using the DSM objective, and

hence solve the following optimization problem:

$$\min_{\theta} \max_{\lambda} \frac{1}{2} \int_0^T \mathbf{E}_{x_0} \{ \mathbf{E}_{x_t|x_0}[\lambda(x_t,t)(\|\nabla_{x_t} \log p_{0t}(x_t|x_0) - \nabla_{x_t} \log p_t(x_t)\|_2^2 \\ + \|s_\theta(x_t,t) - \nabla_{x_t} \log p_t(x_t)\|_2^2)] \} \, \mathrm{d}t. \quad (12)$$

As a result, $\lambda$ is biased by the non-vanishing constant and might not explicitly focus on regions where the model is distant from the true score function. We hypothesize that the denoising score matching approximation can still yield the improvements of an adversarial divergence, and report the empirical evidence to support it in Section 4.2. For simplicity, we limit $\lambda$ to the set of linear functions on $t$, thus removing the dependence on $x$. Details are described in the Appendix in A.8. In order to prevent omission of any values of $t$ in the training, we interpolate between training on the $D_{KL}$ and $D_\lambda$ objective with a hyperparameter $p_{KL}$ which determines the percentage of training on $D_{KL}$.

## 3.2 THE CHOICE OF INITIAL NOISE SCALE

The initial noise scale controls the quality and diversity of the generated samples. It is proposed by Song & Ermon (2020) that the initial noise scale has to be chosen so that the sampling trajectory can traverse from every mode to the other one. Even though this looks good in terms of diversity, it can be quite wasteful because the noise must be large enough so that the whole empirical data distribution is covered. In fact, it is not necessary to traverse from every mode to every other mode directly. We can traverse from a mode to a number of nearest modes and then from those modes to the farther ones. This way, we can choose a significantly smaller initial noise scale that saves us many steps in the sampling trajectory.

According to Song & Ermon (2020), $\sigma(T)$ should be set numerically equal to the maximum pairwise distance of images, which is approximately 50 for CIFAR-10 training images. However, we noticed that smaller values of $\sigma(T)$ are sufficient for generating diverse images. When using smaller initial noise scales, we can further approximate the noise distribution using the sum of gaussian noise $z \sim \mathcal{N}(0, \sigma(T))$ and an additional uniform random variable in the image domain $u \sim U([0,1]^d)$, ensuring the generated images cover the whole image domain. We evaluate qualitative diversity and FID of generated images for various initial noise scales in Section 4.3.

## 4 RESULTS

### 4.1 DIFFUSION-BASED REPRESENTATION LEARNING

For all experiments, we use the same function $\sigma(t), t \in [0,1]$ as in Song et al. (2021b), which is $\sigma(t) = \sigma_{\min} (\sigma_{\max}/\sigma_{\min})^t$, where $\sigma_{\min} = 0.01$ and $\sigma_{\max} = 50$. Further, we use a 2d latent space and set $\lambda(t) = \sigma^2(t)$, which has been shown to yield the KL-Divergence objective (Song et al. (2021a)). Our goal is not to produce state-of-the-art image quality, rather showcase the representation learning method. Because of that and also limited computational resources, we did not carry out an extensive hyperparameter sweep (cf. A.5 for details). Note that all experiments were conducted on a single Tesla V100 GPU, taking up to 30 hours of wall-clock time, which only amounts to 15% of the iterations proposed in Song et al. (2021b). We illustrate how the representation encodes information for denoising in Appendix in Figure 9.

We first train a DRL model with L1-regularization on the latent code on MNIST (LeCun & Cortes (2010)) and CIFAR-10. Figure 2 shows samples from a grid over the latent space and a point cloud visualization of the latent values $z = E_\phi(x_0)$. For MNIST, we can see that the value of $z_1$ controls the stroke width, while $z_2$ weakly indicates the class. The latent code of CIFAR-10 samples mostly encodes information about the background color, which is weakly correlated to the class. The use of a probabilistic encoder (VDRL) leads to similar representations (cf. 5, 6). We further want to point out that the generative process using the reverse SDE involves randomness and thus generates different samples for a single latent representation. The diversity of samples however steadily decreases with the dimensionality of the latent space, which is empirically shown in Figure 10 of the Appendix.

Next, we analyze the behavior of the representation when adjusting the weighting function $\lambda(t)$ to focus on higher noise levels, which can be done by changing the sampling distribution of $t$. To this end, we sample $t \in [0,1]$ such that $\sigma(t)$ is uniformly sampled from the interval $[\sigma_{\min}, \sigma_{\max}] = [0.01, 50]$.

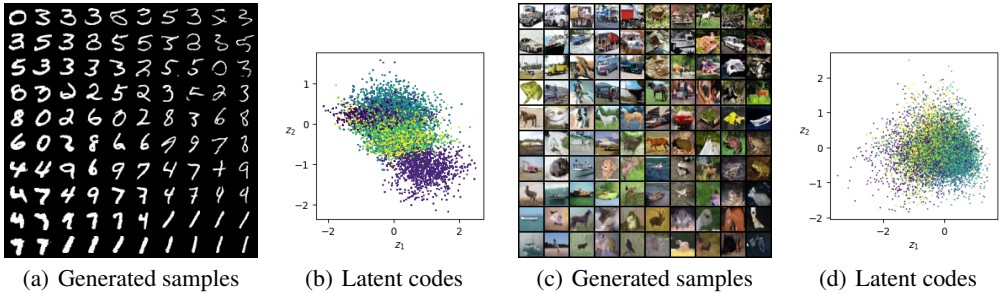

(a) Generated samples     (b) Latent codes     (c) Generated samples     (d) Latent codes

Figure 2: Results of a DRL model trained on MNIST (a-b) and CIFAR-10 (c-d) using uniform sampling of $t$. Samples are generated from a grid of latent values ranging from $-1$ to $1$. The point clouds visualize the latent representation of test samples, colored according to the digit class.

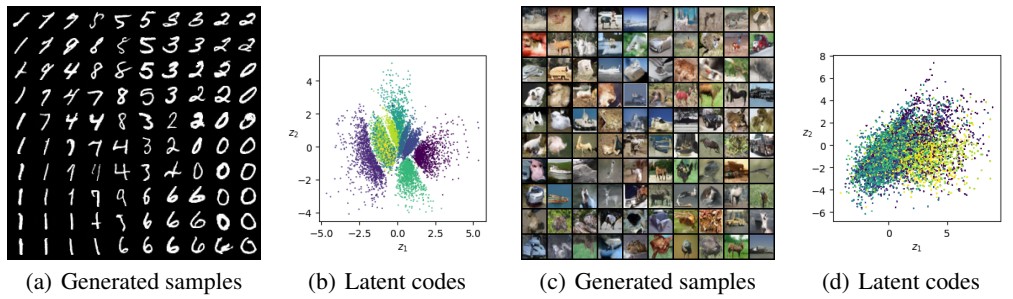

(a) Generated samples     (b) Latent codes     (c) Generated samples     (d) Latent codes

Figure 3: Results of a VDRL model trained with a focus on high noise levels on MNIST (a-b) and CIFAR-10 (c-d). Samples are generated from a grid of latent values ranging from $-2$ to $2$. The point clouds visualize the latent representation of test samples, colored according to the digit class.

Figure 3 shows the resulting representation of VDRL (cf. Fig. 7, 8 for DRL results). As expected, the latent representation for MNIST encodes information about classes rather than fine-grained features such as stroke width. This validates our hypothesis of Section 2.3 that we can control the granularity of features encoded in the latent space. For CIFAR-10, the model again only encodes information about the background, which contains most information about the image. A detailed analysis of class separation in the extreme case of training on single timescales is included in the Appendix (A.4).

Overall, the difference in the latent codes for varying $\lambda(t)$ shows that we can control the granularity encoded in the representation of DRL. While this ability does not exist in previously proposed models for representation learning, it provides a significant advantage when there exists some prior information about the level of detail that we intend to encode in the target representation.

### 4.1.1 APPLICATION TO SEMI-SUPERVISED IMAGE CLASSIFICATION

In the following, we evaluate the infinite-dimensional representation $(E_\phi(x_0, t))_{t \in [0, T]}$ on semi-supervised image classification, where we use DRL and VDRL as pretraining for the LaplaceNet classifier. Table 1 depicts the classifier accuracy on test data for different pretraining settings. Details for architecture and hyperparameters are described in A.7. Note that we do not apply any of the methods proposed in Section 3.

Our proposed pretraining using DRL significantly improves the baseline and often surpasses the state-of-the-art performance of LaplaceNet. Most notable are the results of DRL and VDRL without mixup, which achieve high accuracies without being specifically tailored to the downstream task of classification. Note that pretraining the classifier as part of an autoencoder did not yield any improvements (cf. 6). Combining DRL with mixup yields inconsistent improvements, results are reported in Table 7. In addition, DRL pretraining achieves much better performances when only limited computational resources are available (cf. 4, 5).

We further evaluate the infinite-dimensional representation on few-shot image classification using the representation at different timescales as input. The detailed results are shown in the Appendix (A.6).

| Pretraining
Mixup | | LaplaceNet
None
No | LaplaceNet
None
Yes | Ours
DRL
No | Ours
DRL
Yes | Ours
VDRL
No |
|---|---|---|---|---|---|---|
| Dataset | #labels | | | | | |
| CIFAR-10 | 100 | 73.68 | 75.29 | 74.31 | 64.67 | **81.63** |
| | 500 | 91.31 | 92.53 | **92.70** | 92.31 | **92.79** |
| | 1000 | 92.59 | 93.13 | **93.24** | **93.42** | **93.60** |
| | 2000 | 94.00 | 93.96 | **94.18** | 93.91 | 93.96 |
| | 4000 | 94.73 | 94.97 | 94.75 | **95.22** | **95.00** |
| CIFAR-100 | 1000 | 55.58 | 55.24 | **55.85** | 55.74 | **56.47** |
| | 4000 | 67.07 | 67.25 | 67.22 | **67.47** | **67.54** |
| | 10000 | 73.19 | 72.84 | **73.31** | **73.66** | **73.50** |
| | 20000 | 75.80 | 76.07 | **76.46** | **76.88** | **76.64** |
| MiniImageNet | 4000 | 58.40 | 58.84 | **58.95** | **59.29** | **59.14** |
| | 10000 | 66.65 | 66.80 | **67.31** | 66.63 | **67.46** |

Table 1: Comparison of classifier accuracy in % for different pretraining settings. Scores better than the SOTA model (LaplaceNet) are in **bold**. "DRL" pretraining is our proposed representation learning, and "VDRL" the respective version which uses a probabilistic encoder.

| $p_{KL}$ | 0 | 0.05 | 0.5 | 0.95 | 1 |
|---|---|---|---|---|---|
| FID $\downarrow$ | 3.40 | 3.36 | 3.33 | 3.37 | 3.62 |
| IS $\uparrow$ | 9.73 | 9.77 | 9.79 | 9.50 | 9.44 |

Table 2: FID and Inception Score for different interpolations between maximum likelihood training and training based on the adversarial $\lambda'$. $p_{KL} = 1.0$ corresponds to original training.

In summary, the representations of DRL and VDRL achieve significant improvements compared to that of an autoencoder or VAE for several values of $t$ and are similar for $t$ close to 1.

Overall the results align with the theoretical foundation in Proposition 1 that the rich representation of DRL is at least as good as the static code learned using a reconstruction objective. It further shows that in practice, the infinite-dimensional code is superior to the static representation for the application to downstream tasks by a significant margin.

## 4.2 ADVERSARIAL TRAINING IN DIFFUSION-BASED GENERATIVE MODELS

We evaluate our approach of optimizing the adversarial $\lambda$-Divergence on the task of synthetic image generation of CIFAR-10 images. We evaluate our approach for different values for $p_{KL}$, which is used to interpolate between the original and adversarial training objective. The resulting FID and Inception Scores are displayed in Table 2. The results show that for any value of $p_{KL} \leq 0.95$, the sample quality is improved significantly. However in contrast to our intentions, the adversary converges to the extreme value of $\alpha \approx -1$ in the first 10k iterations and does not change afterwards (cf. 14). While our adversary learns to put focus on values of $t$ where the model is bad, the model apparently cannot improve in this region (cf. 13).

Overall, the higher image quality after training with the adversary indicates that diffusion models can be improved using adversarial training. Since the loss is not significantly reduced, more complex adversaries as in Hardy et al. (2018) will only have the same effect as a predefined $\lambda$. As shown in our experiments, there is much room for improvement by searching for a good $\lambda$. Furthermore, additional improvements might be achieved in the future by introducing inductive biases allowing the model to capture more high-frequency information, thus being able to reduce the loss for small values of $t$.

## 4.3 THE CHOICE OF INITIAL NOISE SCALE

In the following, we evaluate image quality and diversity for different initial noise scales. Note that we do not change $\sigma(T)$, but instead evaluate generated images for different initial times $t_{init}$, which

| $t_{init}$ | $\sigma(t_{init})$ | Gaussian FID ↓ | Uniform + Gaussian FID ↓ |
|---|---|---|---|
| 0.5 | 0.71 | 218.95 | 25.02 |
| 0.6 | 1.66 | 75.11 | 5.15 |
| 0.7 | 3.88 | 12.57 | 2.98 |
| 0.8 | 9.10 | 3.05 | 2.99 |
| 0.9 | 21.33 | 2.97 | 2.94 |
| 1.0 | 50.00 | 3.01 | 2.99 |

Table 3: FID for different initial noise scales evaluated on 20k generated samples.

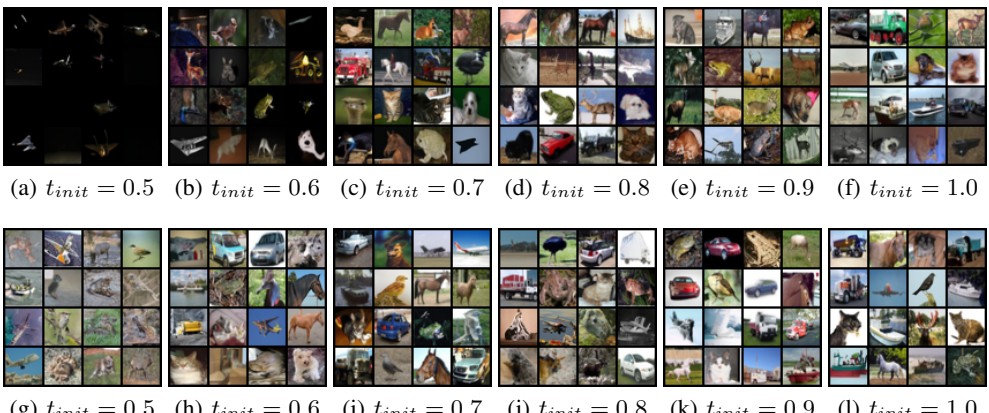

(a) $t_{init} = 0.5$  (b) $t_{init} = 0.6$  (c) $t_{init} = 0.7$  (d) $t_{init} = 0.8$  (e) $t_{init} = 0.9$  (f) $t_{init} = 1.0$

(g) $t_{init} = 0.5$  (h) $t_{init} = 0.6$  (i) $t_{init} = 0.7$  (j) $t_{init} = 0.8$  (k) $t_{init} = 0.9$  (l) $t_{init} = 1.0$

Figure 4: Generated image samples for different values of $t_{init}$. Top row ((a)-(f)) uses the gaussian prior, bottom row ((g)-(l)) uses the version with an additional uniform random variable in the prior.

implicitly define the initial noise scale $\sigma(t_{init})$. This reduces the number of sampling steps per image, which is $1000 \cdot t_{init}$ and thus directly proportional to $t_{init}$. Table 3 shows the FID of generated images for various values of $t_{init}$. As we can see, the first 200 sampling steps can safely be replaced by approximating the prior directly either with the gaussian or the additional uniform distribution. Interestingly, using the sum of the uniform and gaussian random variables as a prior seed leads to improved image quality. This approximation for $p_{0.7}(x)$ allows us to reduce the number of sampling steps by 30% without sacrificing image quality, which is further supported by the visual quality of generated samples shown in Figure 4. Further, note that FID is occasionally lower for values of $t_{init} < 1.0$ than for $t_{init} = 1$. This suggests that up to these timescales, our prior approximates the distribution better than the diffusion model when starting at $t_{init} = 1.0$.

## 5 CONCLUSION

We presented Diffusion-based Representation Learning (DRL), a new objective for representation learning based on conditional denoising score matching. In doing so, we turned the original non-vanishing objective function into one that can be reduced arbitrarily close to zero by the learned representation. We showed that the proposed method learns interpretable features in the latent space. In contrast to previous approaches, denoising score matching as a foundation comes with the ability to manually control the granularity of features encoded in the representation. We demonstrated that the encoder can learn to separate classes when focusing on high noise levels and encodes fine-grained features such as stroke-width when mainly trained on low level noise. In addition, we proposed an infinite-dimensional representation and demonstrated its effectiveness for downstream tasks such as few-shot classification. Using the representation learning as pretraining for a classifier, we were able to improve the results of LaplaceNet, a state-of-the-art model on semi-supervised image classification. As side-contributions, we further showed how adversarial training in diffusion-based models can improve sample quality and were able to increase sampling speed using an approximation of the density at smaller noise scales.

## 6 REPRODUCIBILITY STATEMENT

In order to ensure reproducibility, the code used to run all experiments is attached as supplementary material and will be published based upon acceptance.

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

# A  APPENDIX

## A.1  RELATED WORK ON CONTRASTIVE LEARNING

The core idea of contrastive learning is to learn representations that are similar for different views of the same image and distant for different images. In order to prevent the collapse of representations to a constant, various approaches have been introduced. SimCLRv2 directly includes a loss term repulsing negative image pairs in addition to the attraction of different views of positive pairs (Chen et al. (2020c)). In contrast, BYOL relies solely on positive pairs, preventing collapse by enforcing similarity between the encoded representation of an image and the output of a momentum encoder applied to a different view of the same image (Grill et al. (2020)). An additional approach relies on online clustering and was proposed in SwAV (Caron et al. (2021)). Training in SwAV is based on enforcing consistency between cluster assignments produced for different views of an image. Each of these methods rely on the foundation of Siamese networks (Bromley et al. (1993)), which were shown to be competitive for unsupervised pretraining for classification networks on its own when including a stop-gradient operation on one of the branches (Chen & He (2020)).

## A.2  DENOISING SCORE MATCHING

The following is the proof for the new formulation of the denoising score matching objective in equation 5.

*Proof.* It was shown by Vincent (2011) that equation 4 is equal to explicit score matching up to a constant which is independent of $\theta$, that is,

$$\mathbf{E}_{x_0}\{\mathbf{E}_{x_t|x_0}[\|s_\theta(x_t,t) - \nabla_{x_t}\log p_{0t}(x_t|x_0)\|_2^2]\} \tag{13}$$

$$= \mathbf{E}_{x_t}\left[\|s_\theta(x_t,t) - \nabla_{x_t}\log p_t(x_t)\|_2^2\right] + c. \tag{14}$$

As a consequence, the objective is minimized when the model equals the ground-truth score function $s_\theta(x_t,t) = \nabla_x \log p_t(x)$. Hence we have:

$$\mathbf{E}_{x_0}\{\mathbf{E}_{x_t|x_0}[\|\nabla_{x_t}\log p_t(x_t) - \nabla_{x_t}\log p_{0t}(x_t|x_0)\|_2^2]\} \tag{15}$$

$$= \mathbf{E}_{x_t}\left[\|\nabla_{x_t}\log p_t(x_t) - \nabla_{x_t}\log p_t(x_t)\|_2^2\right] + c \tag{16}$$

$$= c. \tag{17}$$

Combining these results leads to the claimed exact formulation of the Denoising Score Matching objective:

$$J_t^{DSM}(\theta) = \mathbf{E}_{x_0}\{\mathbf{E}_{x_t|x_0}[\|s_\theta(x_t,t) - \nabla_{x_t}\log p_{0t}(x_t|x_0)\|_2^2]\} \tag{18}$$

$$= \mathbf{E}_{x_t}\left[\|s_\theta(x_t,t) - \nabla_{x_t}\log p_t(x_t)\|_2^2\right] + c \tag{19}$$

$$= \mathbf{E}_{x_t}\left[\|s_\theta(x_t,t) - \nabla_{x_t}\log p_t(x_t)\|_2^2\right] \\ + \mathbf{E}_{x_0}\{\mathbf{E}_{x_t|x_0}[\|\nabla_{x_t}\log p_t(x_t) - \nabla_{x_t}\log p_{0t}(x_t|x_0)\|_2^2]\} \tag{20}$$

$$= \mathbf{E}_{x_0}\{\mathbf{E}_{x_t|x_0}[\|\nabla_{x_t}\log p_{0t}(x_t|x_0) - \nabla_{x_t}\log p_t(x_t)\|_2^2 \\ + \|s_\theta(x_t,t) - \nabla_{x_t}\log p_t(x_t)\|_2^2]\}. \tag{21}$$

$\square$

## A.3  REPRESENTATION LEARNING

Here we present the proof for Proposition 1, stating that the infinite-dimensional code learned using DRL is at least as good as a static code learned using a reconstruction objective.

*Proof.* We assume that the distribution of the diffused samples at time $t = T$ matches a known prior $p_T(x_T)$. That is, $\int p(x_0)p_{0T}(x_T|x_0)\,\mathrm{d}x_0 = p_T(x_T)$. In practice $T$ is chosen such that this assumption approximately holds.

Now consider the training objective in equation 10 at time $T$, which can be transformed to a reconstruction objective in the following way:

$$\lambda(T)\mathbf{E}_{x_0,x_T}\left[\|s_\theta(x_T,T,E_\phi(x_0,T)) - \nabla_{x_T}\log p_{0T}(x_T|x_0)\|_2^2\right] \tag{22}$$

$$=\lambda(T)\mathbf{E}_{x_0}\mathbf{E}_{x_T\sim p_T(x_T)}\left[\left\|s_\theta(x_T,T,E_\phi(x_0,T)) - \frac{x_0-x_T}{\sigma^2(T)}\right\|_2^2\right] \tag{23}$$

$$=\lambda(T)\sigma^{-4}(T)\mathbf{E}_{x_0}\mathbf{E}_{x_T\sim p_T(x_T)}\left[\|D_\theta(E_\phi(x_0,T)) - x_0\|_2^2\right] \tag{24}$$

$$=\lambda(T)\sigma^{-4}(T)\mathbf{E}_{x_0}\left[\|D_\theta(E_\phi(x_0,T)) - x_0\|_2^2\right], \tag{25}$$

where we replaced the score model with a Decoder model $s_\theta(x_T,T,E_\phi(x_0,T)) = \frac{D_\theta(E_\phi(x_0,T))-x_T}{\sigma^2(T)}$ and replaced the score function of the perturbation kernel $\nabla_{x_T}\log p_{0T}(x_T|x_0)$ with its known closed-form solution $\frac{x_0-x_T}{\sigma^2(T)}$ determined by the Forward SDE in equation 1. Hence the learned code at time $t = T$ is equal to a code learned using a reconstruction objective.

We model a downstream task as a minimization problem of a distance $d : \Omega \times \Omega \to \mathbb{R}$ in the feature space $\Omega$ between the true feature extractor $g : \mathbb{R}^d \to \Omega$ which maps data samples $x_0$ to a features space $\Omega$ and a model feature extractor $h_\psi : \mathbb{R}^c \to \Omega$ doing the same given the code as input. The following shows that the infinite-dimensional representation is at least as good as the static code:

$$\inf_t \min_\psi \mathbf{E}_{x_0}[d(h_\psi(E_\phi(x_0,t)), g(x_0))] \le \min_\psi \mathbf{E}_{x_0}[d(h_\psi(E_\phi(x_0,T)), g(x_0))] \tag{26}$$

$\square$

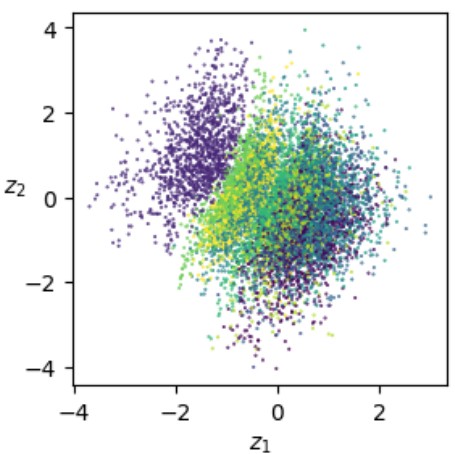

(a) Samples generated from a grid of latent values on a range from $-2$ to $2$

(b) Latent representation of test samples, colored according to the digit class

Figure 5: Samples and latent distribution of a model trained on MNIST using KL-divergence and uniform sampling of $t$

## A.4 TRAINING ON SINGLE TIMESCALES

To understand the effect of training DRL on different timescales more clearly, we limit the support of the weighting function $\lambda(t)$ to a single value of $t$. We analyze the resulting quality of the latent representation for different values of $t$ using the silhouette score with euclidean distance based on the dataset classes Rousseeuw (1987). It compares the average distance between a point to all other points in its cluster with the average distance to points in the nearest different cluster. Thus we measure how well the latent representation encodes classes, ignoring any other features. Note that

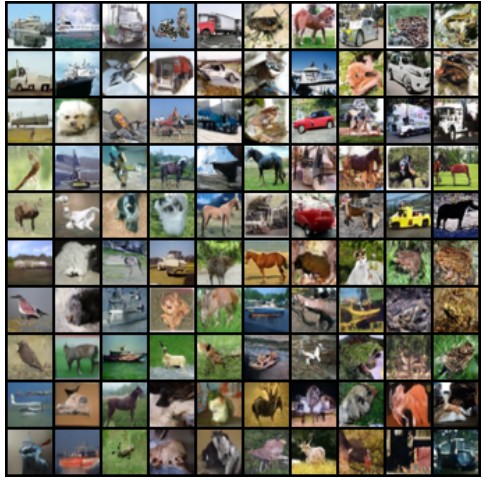 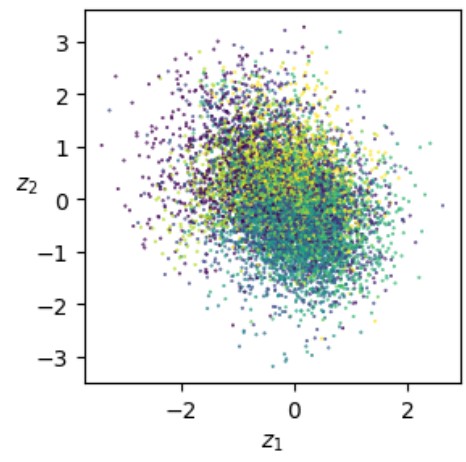

(a) Samples generated from a grid of latent values on a range from $-1$ to $1$

(b) Latent representation of test samples, colored according to the class label

Figure 6: Samples and latent distribution of a VDRL model trained on CIFAR-10 using uniform sampling of $t$.

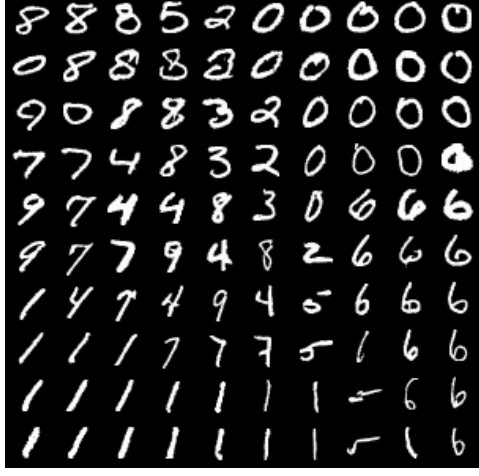 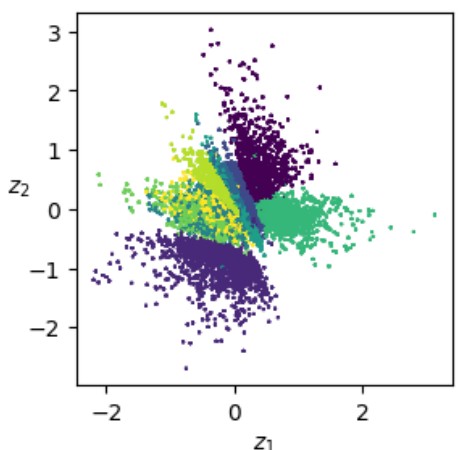

(a) Samples generated from a grid of latent values on a range from $-1$ to $1$

(b) Latent representation of test samples, colored according to the digit class

Figure 7: Samples and latent distribution of a DRL model trained on MNIST using uniform sampling of $\sigma$ (focus on high noise levels).

after learning the representation with a different distribution of $t$ it is necessary to perform additional training with uniform sampling of $t$ and a frozen encoder to achieve good sample quality.

Figure 11 shows the silhouette scores of latent codes of MNIST and CIFAR-10 samples for different values of $t$. In alignment with our hypothesis of Section 2.3, training DRL on a small $t$ and thus low noise levels leads to almost no encoded class information in the latent representation, while the opposite is the case for a range of $t$ which differs between the two datasets. The decline in encoded class information for high values of $t$ can be explained by the vanishing difference between distributions of perturbed samples when $t$ gets large. This shows that the distinction among the code classes represented by the silhouette score is controlled by $\lambda(t)$.

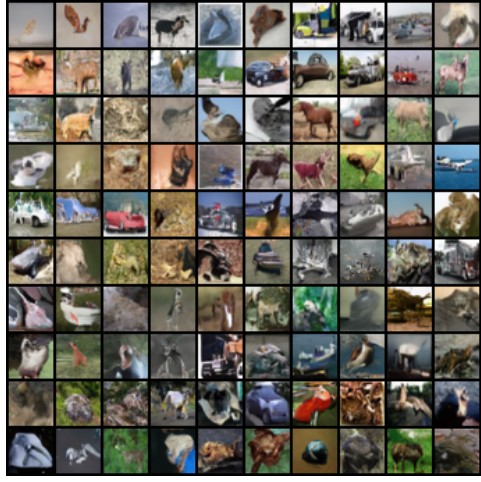 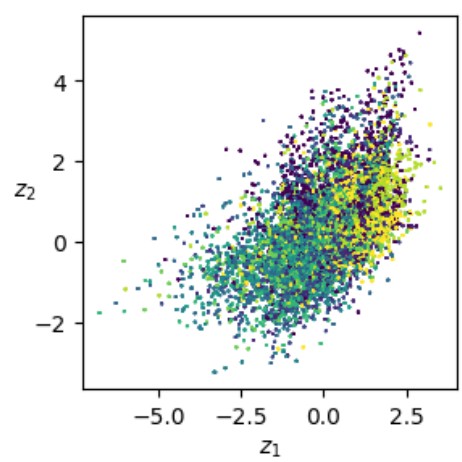

(a) Samples generated from a grid of latent values on a range from −1 to 1

(b) Latent representation of test samples, colored according to the class label

Figure 8: Samples and latent distribution of a DRL model trained on CIFAR-10 using uniform sampling of $\sigma$ (focus on high noise levels).

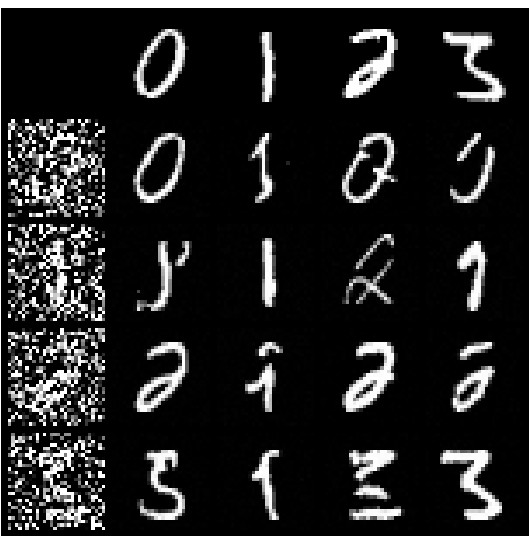

Figure 9: Samples generated starting from $x_t$ (left column) using the diffusion model with the latent code of another $x_0$ (top row) as input. It shows that samples are denoised correctly only when conditioning on the latent code of the corresponding original image $x_0$.

### A.5 ARCHITECTURE AND HYPERPARAMETERS

The model architecture we use for all experiments is based on "DDPM++ cont. (deep)" used for CIFAR-10 in Song et al. (2021b). It is composed of a downsampling and an upsampling block with residual blocks at multiple resolutions. We did not change any of the hyperparameters of the optimizer. Depending on the dataset, we adjusted the number of resolutions, number of channels per resolution, and the number of residual blocks per resolution in order to reduce training time.

For representation learning, we use an encoder with the same architecture as the downsampling block of the model, followed by another three dense layers mapping to a low dimensional latent space. Another four dense layers map the latent code back to a higher-dimensional representation. It is then

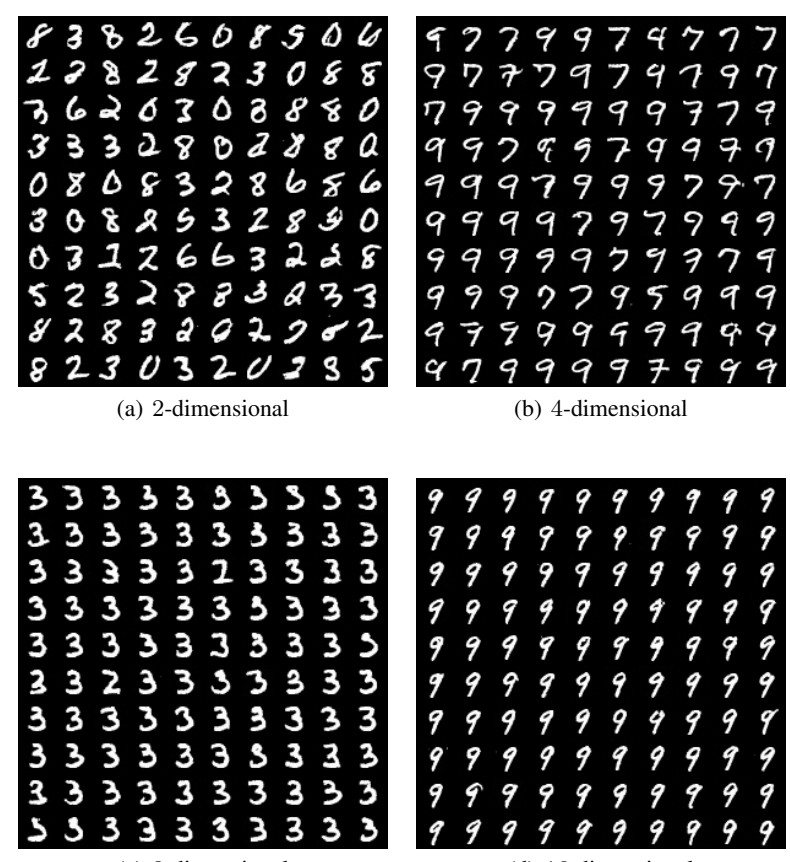

(a) 2-dimensional      (b) 4-dimensional

(c) 8-dimensional      (d) 16-dimensional

Figure 10: Samples generated using the same latent code for each generation, showing that the randomness of the code-conditional generation of DRL reduces in higher dimensional latent spaces.

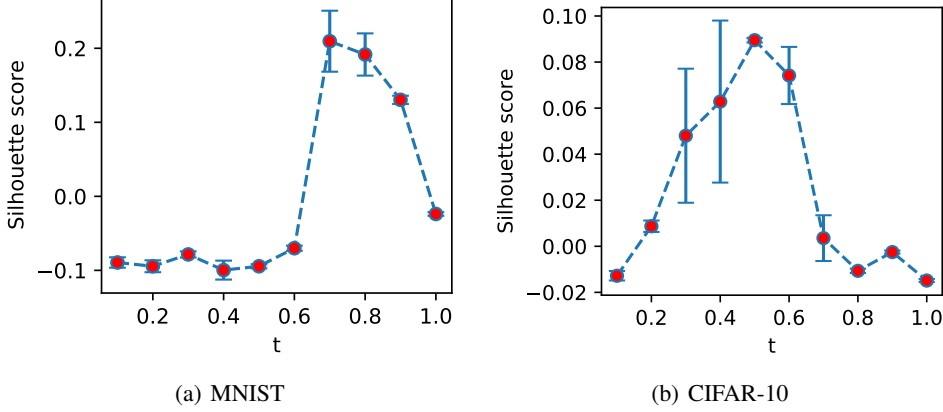

(a) MNIST      (b) CIFAR-10

Figure 11: Mean and standard deviation of silhouette scores when training a DRL model on MNIST (left) and CIFAR-10 (right) using a single $t$ over three runs.

given as input to the model in the same way as the time embedding. That is, each channel is provided with a conditional bias determined by the representation and time embedding at multiple stages of the downsampling and upsampling block.

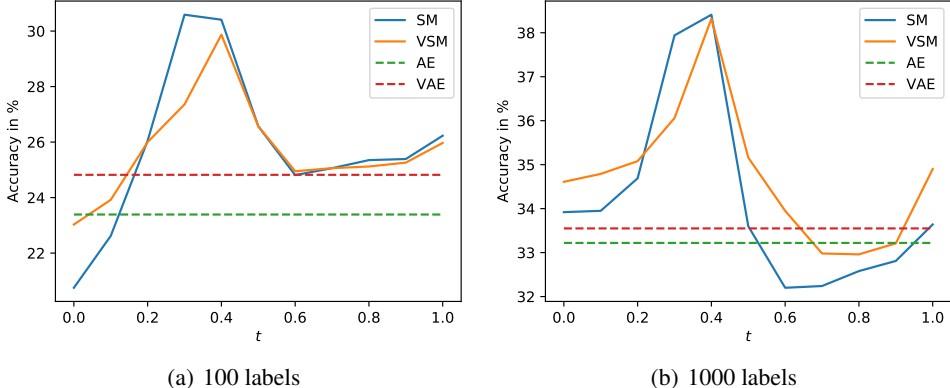

(a) 100 labels           (b) 1000 labels

Figure 12: Classifier accuracies for few shot learning on given 8-dimensional representations learned using DRL (SM), VDRL (VSM), Autoencoder (AE) and Variational Autoencoder (VAE).

**Regularization of the latent space**    For both datasets, we use a regularization weight of $10^{-5}$ when applying L1-regularization, and a weight of $10^{-7}$ when using a probabilistic encoder regularized with KL-Divergence.

**MNIST hyperparameters**    Due to the simplicity of MNIST, we only use two resolutions of size $28 \times 28 \times 32$ and $14 \times 14 \times 64$, respectively. The number of residual blocks at each resolution is set to two. In each experiment, the model is trained for $80k$ iterations. For uniform sampling of $\sigma$ we trained the models for an additional $80k$ iterations with a frozen encoder and uniform sampling of $t$.

**CIFAR-10 hyperparameters**    For the silhouette score analysis, we use three resolutions of size $32 \times 32 \times 32$, $16 \times 16 \times 32$, and $8 \times 8 \times 32$, again with only two residual blocks at each resolution. Each model is trained for $90k$ iterations.

**CIFAR-10 (deep) hyperparameters**    While representation learning works for small models already, sample quality on CIFAR-10 is poor for models of the size described above. Thus for models used to generate samples, we use eight residual blocks per resolution and the following resolutions: $32 \times 32 \times 32$, $16 \times 16 \times 64$, $8 \times 8 \times 64$, and $4 \times 4 \times 64$. Each model is trained for $300k$ iterations. Note that this number of iterations is not sufficient for convergence, however capable of illustrating the representation learning with limited computational resources.

A.6    EVALUATION OF THE INFINITE-DIMENSIONAL REPRESENTATION

In order to evaluate our infinite-dimensional representation, we conduct an ablation study where we compare our proposed method with Autoencoders (AE) and Variational Autoencoders (VAE) on CIFAR-10 images. We measure the accuracy of an SVM provided by sklearn (Pedregosa et al. (2011)) with default hyperparameters trained on the representation of 100 (resp. 1000) training samples and their class labels. For our time-dependent representation, this is done for fixed values of $t$ between $0.0$ and $1.0$ in steps of $0.1$. This is done for both DRL and VDRL, where we use a probabilistic encoder regularized by including an additional KL-Divergence term in the training objective. DRL and AE were regularized using L1-norm, and the regularization weight was optimized for each model independently.

Results for few-shot learning with fixed representations are shown in Figure 12. As expected, the accuracies when training on the score matching representations highly depend on the value of $t$. Overall our representation achieves much better scores when using the best $t$, and performs comparable to AE and VAE for $t = 1.0$. This aligns with Proposition 1 claiming that our representation learning method for $t = 1.0$ is similar to a static code learned using reconstruction objective. Note that the shape of the time-dependent classifier accuracies resemble the one of the silhouette score of CIFAR-10 in 11. This is not surprising, since both training on single values of $t$ and learning a time-dependent representation are both trained to find the optimal representation for a given value of $t$. We further

| Dataset | #labels | No pretraining | Pretraining using DRL | Improvement |
|---------|---------|----------------|----------------------|-------------|
| CIFAR-10 | 100 | 64.12 | 69.79 | +5.67 |
| | 500 | 86.24 | 88.28 | +2.04 |
| | 1000 | 87.48 | 88.56 | +1.08 |
| | 2000 | 89.99 | 89.52 | -0.47 |
| | 4000 | 90.15 | 91.13 | +0.98 |
| CIFAR-100 | 1000 | 45.14 | 48.04 | +2.90 |
| | 4000 | 59.86 | 60.34 | +0.48 |
| | 10000 | 64.83 | 65.80 | +0.97 |
| | 20000 | 65.77 | 66.39 | +0.62 |
| MiniImageNet | 4000 | 47.18 | 50.75 | +3.57 |
| | 10000 | 58.66 | 58.62 | -0.04 |

Table 4: Classifier accuracy in % with and without DRL as pretraining of the classifier when training for 100 epochs only.

want to point out that representation learning through score matching enjoys the training stability of diffusion-based generative models, which is often not the case in GANs and VAEs.

### A.7 SEMI-SUPERVISED IMAGE CLASSIFICATION

**Architecture and Hyperparameters** In all experiments, our encoder has the same architecture as the classifier, where the hidden layer used to measure similarities for assigning pseudo-labels in LaplaceNet is used as the latent code in representation learning. For all experiments, the input $t$ to the encoder is included as a trainable parameter of the model and initialized with $t = 0.5$. As done in the original paper, we train the model for 260 iterations, where each iteration consists of assigning pseudo-labels and one epoch of supervised training on the assigned pseudo-labels. The training is preceded by 100 supervised epochs on the labeled data. We use the small WideResNet model WRN-28-2 of Sellars et al. (2021) and the same hyperparameters as the authors.

**Evaluation with limited computation time** In the following we include a more detailed analysis on the scenario of few supervised labels and limited computational resources. Besides LaplaceNet and its version without mixup, we include an ablation study of encoder pretraining as part of an autoencoder using binary cross entropy as reconstruction objective. In addition, we propose to improve the search for the optimal value of $t$ by model selection, since the gradient for $t$ is usually noisy and small. Thus we include additional experiments where we chose the initial $t$ based on the minimum training loss after 100 epochs of supervised training. The optimal $t$ is approximated by calculating the training loss for 11 equally spaced values of $t$ in the interval $[0.001, 1]$. The results are shown in Table 5. While mixup achieves no significant improvement in the few-label case trained using 100 epochs, we can see that a simple autoencoder pretraining consistently improves classifier accuracy. More notably however, our proposed pretraining based on score matching achieves significantly better results than both random initialization and autoencoder pretraining. In the $t$-search, we observed that for all datasets, our proposed method selects $t = 0.9$, however it moves towards the interval $[0.4, 0.6]$ during training. While this shows that our the approach of selecting $t$ based on supervised training loss is not working, it demonstrates that the parameter $t$ can very well be learned in the training process, making the downstream task performance robust to the initial value of $t$. In our experiments the final value of $t$ was always in the range $[0.4, 0.6]$, independent of the initial value of $t$.

| Pretraining | Options | CIFAR-10 100 labels | CIFAR-100 1000 labels | MiniImageNet 4000 labels |
|---|---|---|---|---|
| None | | 64.12 | 45.14 | 47.18 |
| None | mixup | 54.06 | 46.28 | 47.64 |
| DRL | | **69.79** | **48.04** | **50.75** |
| DRL | $t$-search | 67.07 | 47.08 | 50.31 |
| Autoencoder | | 64.99 | 46.88 | 48.52 |

Table 5: Comparison of classifier accuracy in % for different pretraining methods in the case of few supervised labels when training for 100 epochs only.

| Pretraining | CIFAR-10 100 labels | CIFAR-100 1000 labels | MiniImageNet 4000 labels |
|---|---|---|---|
| None | 73.68 | 55.58 | 58.40 |
| DRL | **74.31** | **55.85** | **58.95** |
| Autoencoder | 58.84 | 55.41 | 57.93 |

Table 6: Classifier accuracy in % for autoencoder pretraining compared with the baseline and score matching as pretraining. No mixup is applied for this ablation study.

| Pretraining Mixup in sup. training | | Ours Basic No | Ours Basic Yes | Ours Mixup-DRL Yes | Ours VDRL No | Ours VDRL Yes |
|---|---|---|---|---|---|---|
| Dataset | #labels | | | | | |
| CIFAR-10 | 100 | 74.31 | 64.67 | 70.40 | **81.63** | 77.51 |
| | 500 | 92.70 | 92.31 | 92.55 | **92.79** | 91.46 |
| | 1000 | 93.24 | 93.42 | 93.14 | **93.60** | 93.33 |
| | 2000 | **94.18** | 93.91 | 93.80 | 93.96 | 94.27 |
| | 4000 | 94.75 | **95.22** | 94.75 | 95.00 | 94.87 |
| CIFAR-100 | 1000 | 55.85 | 55.74 | 55.15 | **56.47** | 55.65 |
| | 4000 | 67.22 | 67.47 | 67.09 | **67.54** | 67.52 |
| | 10000 | 73.31 | 73.66 | **74.36** | 73.50 | 73.20 |
| | 20000 | 76.46 | 76.88 | **77.04** | 76.64 | 76.68 |
| MiniImageNet | 4000 | 58.95 | 59.29 | **59.46** | 59.14 | 59.36 |
| | 10000 | 67.31 | 66.63 | 67.31 | **67.46** | 66.79 |

Table 7: Evaluation of classifier accuracy in %, including the setting of using mixup during pretraining (right column). DRL pretraining is our proposed representation learning, and "Mixup-DRL" the respective version which additionally applies mixup during pretraining. "VDRL" instead uses a probabilistic encoder.

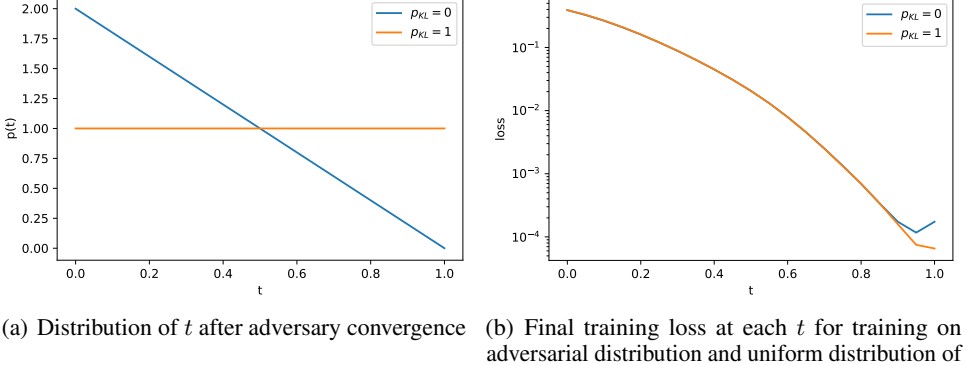

(a) Distribution of $t$ after adversary convergence   (b) Final training loss at each $t$ for training on adversarial distribution and uniform distribution of $t$

Figure 13: Comparison of the distribution of $t$ and the respective loss with and without including the adversary $\lambda'_\alpha$.

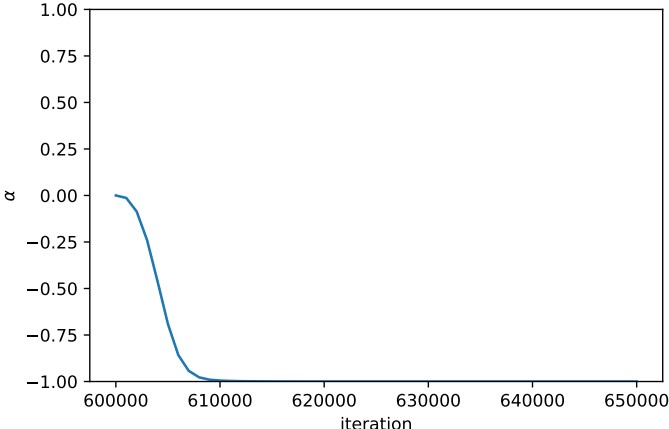

Figure 14: Value of parameter $\alpha$ over training, where $2\alpha$ is the slope of the adversary $\lambda'_\alpha$.

## A.8 Adversarial training

For evaluating adversarial training, we initialize our model using a pretrained checkpoint which has been trained for 600k iterations. We then alternate training iterations between the model and the adversary, which is done for an overall number of 50k iterations.

In order to prevent $\lambda$ from increasing indefinitely, we propose to fix it to a certain function class. For simplicity, one can remove $\lambda$'s dependence on $x$, thus setting $\lambda(t) = \lambda(x, t)$. It was shown by Song et al. (2021b) that setting $\lambda(t) = \sigma^2(t)$ yields the KL-Divergence objective $D_{KL}$. Hence we chose $\lambda(t) = \lambda'(t)\sigma^2(t)$ to have KL-Divergence as the initial divergence when $\lambda'(t) = 1 \, \forall t \in [0, T]$. We further propose to limit $\lambda'$ to the set of linear functions with slopes ranging from $-1$ to $1$. Formally, we set $\lambda'_\alpha(t) = 2\alpha t + (1 - \alpha), \alpha \in [-1, 1]$. Note that using this formulation, $\lambda'_\alpha(t)$ can be seen as a probability distribution over $t \in [0, T]$. In addition, we noticed that in practice sampling $t$ according to $\lambda'(t)$ instead of weighting the loss consistently achieves better results and thus used this approach in our evaluation in Section 4.2.

