# OpenReview forum: "Diffusion-Based Representation Learning"
_ICLR.cc/2022/Conference — ICLR 2022 Submitted_

### Official Review · Reviewer_Wb9v · 2021-10-25

**Correctness:** 4
**Technical Novelty And Significance:** 3
**Empirical Novelty And Significance:** 3
**Recommendation:** 6
**Confidence:** 3

**Main Review:**

Overall, the paper is well written and the method seems sound. The topic of score-based generative models as an alternative to GANs and VAEs is worth exploring and the authors propose an interesting way to generate powerful representations.

The fact that pretraining with such method allows to improve the state-of-the-art in the semi-supervised classification setting is an important contribution, even though the improvement seems less significant on datasets with more classes than CIFAR-10. It would also be desirable to have standard deviations associated to the classification accuracy values to better assess statistical significance of the result.

Results on the generation task do not seem particularly impressive but it is fine for it not to be the main focus.

A couple of typos should be fixed: page 1 the acronym SDE has not yet been defined; page 3 Densoising --> Denoising.

**Summary Of The Paper:**

The paper presents a novel technique for representation learning by means of generative models. In particular, non-adversarial score-matching methods are improved to control the level of detail in the representations and learn a infinite-dimensional code. This is particularly useful for the semi-supervised learning setting, where it is shown that pretraining via the proposed model allows strong class separation and leads to state of the art results.

**Summary Of The Review:**

Pros:
- Interesting extension of current non-adversarial generative models, with expanded theoretical results
- Powerful representation learning capabilities and SoTA on semi-supervised classification
- Adequate experimental assessment and ablation
Cons:
- Results on image generation are not very impressive
- Statistical significance of the classification results could be better assessed

I think that overall the paper is a valuable contribution and expands the knowledge on representation learning techniques in a meaningful way.

---

> ### Author Response · Authors · 2021-11-16
> **Authors' response to Reviewer Wb9v**
>
> We thank the respected reviewer for the valuable comments. We appreciate the positive feedback that recognizes the contributions of our work. We address the major comments here and take into account the other comments on writing in the revised manuscript.
> RCy: Reviewer’s concern No.y
> AR: Authors’ response
>
> RC1. Results on image generation are not very impressive
>
> AR: We agree with the respected reviewer that the results of the image generation task are not so impressive compared to SOTA that is optimized for image quality. However, we would like to emphasize that the focus of this work was to show a natural extension of score-based generative models to representation learning rather than improving their image generation quality. Due to computational constraints, we could not achieve the same quality of image generation even by running the original codes as we could not afford the same amount of compute.
>
> RC2. Statistical significance of the classification results could be better assessed
>
> AR: We totally agree with the respected reviewer that statistical significance obtained by many runs could make assessment much better. Due to the computational budget and the cost of training score-based generative models, we managed to run only a few experiments multiple times and did not observe significant and decisive randomness in the results. The limited number of runs with different seeds however allowed for the evaluation of the approach for multiple datasets and various numbers of labels.

---

> > ### Comment · Reviewer_Wb9v · 2021-11-22
> > **Updated comments**
> >
> > After reading the other reviewers' comments and the authors response, I think that recent self-supervised contrastive learning techniques can be considered relevant for this work. Since the authors main experimental result is on classification, they need to contextualize it with respect to those methods to have a fuller picture. I still think there is merit in the theoretical aspects of the paper, but I lowered my score according to these problems in the experimental assessment.

---

> ### Author Response · Authors · 2021-11-21
> **Update on concerns**
>
> Please let us know if our responses answered your concerns and convinced you to improve your initial score. We are more than happy to elaborate more on any point that may still be unclear.

---

### Official Review · Reviewer_pE4h · 2021-10-27

**Correctness:** 4
**Technical Novelty And Significance:** 4
**Empirical Novelty And Significance:** 2
**Recommendation:** 5
**Confidence:** 5

**Main Review:**

1. Strengths.

    The idea, presented in this paper, is novel.  The motivation behind the method, taking its origin in the rethought conditional score matching objective, is clearly explained.


2. Weaknesses.

    a. Firstly, the submitted manuscript misses the detailed review of related works on representation learning. Therefore, it is rather difficult to realize which context the authors put their work in. The restriction of limited computational resources was mentioned, but it is still unclear what the formal restriction is.

    b. In the absence of context, it is difficult to understand the choice of a baseline for the quantitative evaluation. The last two years were very successful for representation learning in the domain of images. Such methods as SimSiam [2], SimCLRv2 [3], BYOL [4] et cetera may be considered as reliable models for comparison. However, they are not even mentioned in the paper. Also, the evaluation protocol used in those works could be applied to the proposed DRL approach.

    c. Though the idea of infinite-dimensional representations looks appealing, I did not understand how it can be applied for the downstream tasks. During the experiment, described in Sec. A5, the authors chose the specific timestamp according to the best classifier accuracy. Thus, the final representation is still point-based. Am I right, that for any other task, e.g. image retrieval, one should again test several values of $t$ and choose the best one?


3. Questions and suggestions.

    a. It is claimed in Sec. 3.1. that $\lambda$-divergences can express any $f$-divergence. However, as far as I can understand, paper [5] has shown that $\lambda$-divergence can bound the KL-divergence. Could you please put the proof sketch or the reference to one of the previous results?

    b. What is the motivation behind the sum of a Uniform r.v. and Gaussian r.v. (Sec. 4.3)?

    c. In Eq. 23 the decoder should be defined as $D_\theta \left(E_\phi \left(x_0, T\right)\right)$ instead of $D_\theta \left(x_T, E_\phi \left(x_0, T\right)\right)$.


4. References

    [2] Chen et al. Exploring Simple Siamese Representation Learning. 2020.

    [3] Chen et al. Big Self-Supervised Models are Strong Semi-Supervised Learners. 2020.

    [4] Grill et al. Bootstrap your own latent: A new approach to self-supervised Learning. 2020.

    [5] Song et al.  Maximum likelihood training of score-based diffusion models. 2021.

**Summary Of The Paper:**

In this paper, the authors propose a new scheme for training representations using diffusion probability models (DDPM). In detail, the estimator $s_\theta$ receives the encoded original image $E_\phi \left(x_0\right)$ (or $E_\phi \left(x_0, t\right)$) as additional input - the DDPM loss function causes the $E_\phi$ encoder to extract the features needed for image reconstruction. Remarkably, this representation can be deterministic or stochastic, pointwise or infinite-dimensional (with time variable $t$).

Qualitative experiments on MNIST and CIFAR-10 show that the encoder can learn some meaningful features. A quantitative comparison with LaplaceNet [1] on CIFAR-10, CIFAR-100, and MiniImageNet, demonstrates an improvement in the quality of semi-supervised classification.

[1] Sellars et al. LaplaceNet: A Hybrid Energy-Neural Model for Deep Semi-Supervised Classification. 2021.

**Summary Of The Review:**

1. Pre-rebuttal rating.

    Despite I admit the idea, presented in the paper, is interesting and novel, the evaluation does not seem conclusive for me. I ask the authors to motivate the choice of the baseline and explain what the use case of the presented approach is. Currently, I tend to rate the submission below the acceptance threshold.

2. Post-rebuttal update

    Having read the authors' feedback and other reviews, I keep the initial rating. While the presented ideas may be of some interest to the community, the evaluation of the approach should be more thorough.

---

> ### Author Response · Authors · 2021-11-16
> **Authors' response to Reviewer pE4h**
>
> We thank the respected reviewer for the valuable comments. We appreciate the positive feedback that recognizes the contributions of our work. We address the major comments here and take into account the other comments on writing in the revised manuscript.
> RCy: Reviewer’s concern No.y
> AR: Authors’ response
>
> RC1. Firstly, the submitted manuscript misses the detailed review of related works on representation learning. Therefore, it is rather difficult to realize which context the authors put their work in. The restriction of limited computational resources was mentioned, but it is still unclear what the formal restriction is.
>
> AR: We thank the reviewer for pointing out the missing discussion of related works and unclear restriction of computational resources. All experiments were conducted on a single Tesla V100 GPU, taking up to 30 hours of wall-clock time. With these resources, we were only able to run 15% of the iterations proposed by Song et al. 2021a. We added this information at the beginning of Section 4.1 of the revised manuscript. In addition, we extended our discussion of related work on representation learning and included a summary of contrastive learning methods in A.1 of the revised manuscript.
>
>
> RC2. In the absence of context, it is difficult to understand the choice of a baseline for the quantitative evaluation. Methods as SimSiam [2], SimCLRv2 [3], BYOL [4] et cetera may be considered as reliable models for comparison. However, they are not even mentioned in the paper. Also, the evaluation protocol used in those works could be applied to the proposed DRL approach.
>
> AR: We thank the reviewer for pointing out the missing context and agree that significant work in representation learning is relevant to our work. We added a paragraph pointing to significant contrastive learning methods in the introduction and an additional summary in A.1 of the revised manuscript. In contrast to contrastive representation learning methods DRL is not specifically designed for augmentation-invariant downstream tasks such as classification and is thus not competitive when evaluating the class separation of the representations. We added this clarification in the introduction to explain the context of our work being the extension of diffusion-based models to representation learning and showcasing improvement opportunities by applying adversarial training and different initial noise scales.
>
>
> RC3. Though the idea of infinite-dimensional representations looks appealing, I did not understand how it can be applied for the downstream tasks. During the experiment, described in Sec. A5, the authors chose the specific timestamp according to the best classifier accuracy. Thus, the final representation is still point-based. Am I right, that for any other task, e.g. image retrieval, one should again test several values of t and choose the best one?
>
> AR: Thanks for recognizing the idea of infinite-dimensional representations as appealing.  We agree with the reviewer that the performance of a method based on the representation is highly dependent on the chosen noise level, thus the best noise level has to be found independently for each downstream task. Theoretically speaking, the downstream task does not have to depend on the representation at a single noise scale. Any function that aggregates representations corresponding to different noise levels can be used. The max function is a simple and intuitive aggregation that chooses only one noise level but more general aggregations are possible as well. We added the respective clarification to Section 2.4.
>
>
> RC4. It is claimed in Sec. 3.1. that λ-divergences can express any f-divergence. However, as far as I can understand, paper [5] has shown that λ-divergence can bound the KL-divergence. Could you please put the proof sketch or the reference to one of the previous results?
>
> AR: The exact equivalence between score matching and maximum-likelihood training only holds if both the diffused distribution at time T is equal to our prior ($p_T = \pi$), and the model is the score function of a valid distribution q_t ($s_θ(x, t) = ∇_x \log q_t(x)$), as shown in Theorem 2 of version 4 of Song et al. 2021a. Since the relation of λ-divergences to f-divergences is only explained in version 1 of Song et al. 2021a, we now adapted the citation to refer to version 1.
>
>
> RC5. What is the motivation behind the sum of a Uniform r.v. and Gaussian r.v. (Sec. 4.3)?
>
> AR: Thanks for pointing out the missing motivation, we added a clarification in Section 3.2 that the uniform r.v. ensures that generated images cover the whole image domain. If the initial noise level is too low, only sampling from the gaussian around zero reduces diversity, whereas additional samples taken from a uniform r.v. ensure the tail of the Gaussian is also covered.
>
>
> RC6. In Eq. 23 the decoder should be defined as Dθ(Eϕ(x0,T)) instead of Dθ(xT,Eϕ(x0,T))
>
> AR: Certainly, thanks for noticing! We corrected it in the revised manuscript.

---

> > ### Comment · Reviewer_pE4h · 2021-11-22
> > **Augmentation-invariant baselines**
> >
> > > In contrast to contrastive representation learning methods DRL is not specifically designed for augmentation-invariant downstream tasks such as classification and is thus not competitive when evaluating the class separation of the representations.
> >
> > However, most of the experiments described in the paper are related to classification problems. Therefore, currently, the submitted manuscript does not prove that the presented approach outperforms the contrastive training-related baselines on any task. To support the cited claim, it would be nice to demonstrate such a problem, that DRL solves better than contrastive baselines do.
> >
> > Another possible advantage of the presented approach is, probably, the speed of training. However, the paper does not show the results of contrastive approaches trained with the same computational budget.

---

> ### Author Response · Authors · 2021-11-21
> **Update on concerns**
>
> Please let us know if our responses answered your concerns and convinced you to improve your initial score. We are more than happy to elaborate more on any point that may still be unclear.

---

### Official Review · Reviewer_VdM1 · 2021-11-08

**Correctness:** 2
**Technical Novelty And Significance:** 2
**Empirical Novelty And Significance:** 3
**Recommendation:** 5
**Confidence:** 4

**Main Review:**

## Strengths

The method proposed in this paper is simple and easy to use. Experimental results also demonstrate improvement over baselines.

## Weaknesses

1. The idea of representation learning with multi-scale denoising score matching is not new. Very similar ideas have already been discussed in at least [2][3][4].

1. Using "diffusion-based" and "score-based" interchangeably in the paper makes it very confusing. I highly recommend sticking to one name. Given the dependency of the proposed approach on denoising score matching losses, I feel "score-based" might be a more appropriate naming.

2. Proposition 1 is quite straightforward. Everyone who understands the proof of denoising score matching should know clearly that it is equivalent to the Fisher divergence up to a constant. It feels inappropriate to write down the expression of the constant as if it were your own contribution.

3. The second sentence in section 3 is misleading. The work of Song et al. 2021a didn't claim diffusion models are trained to minimize KL-divergence—it is only true for a particular weighting of the score matching losses called the likelihood weighting. Also, the statement holds for the likelihood weighting even if the score model is not curl-free.

4. In the second paragraph of section 3.2, authors claimed that the maximum pairwise distance of images is 170 for CIFAR-10. This is incorrect according to song and ermon 2020, where the maximum pairwise distance is reported to be around 50 for images with pixel values in [0, 1].

5. The adversarial training of lambda divergences is not principled because the unknown constant of denoising score matching can depend on lambda. Also authors should cite and discuss the related work [1] which discusses adversarial training for score matching losses.

## References
[1] Jolicoeur-Martineau, Alexia, et al. "Adversarial score matching and improved sampling for image generation." ICLR 2021.

[2] K. J. Geras and C. Sutton. Scheduled denoising autoencoders. arXiv preprint arXiv:1406.3269,
2014.

[3] B. Chandra and R. K. Sharma. Adaptive noise schedule for denoising autoencoder. NeurIPS 2014.

[4] Q. Zhang and L. Zhang. Convolutional adaptive denoising autoencoders for hierarchical feature
extraction. Frontiers of Computer Science, 2018.




**Summary Of The Paper:**

This paper proposes to leverage the recently proposed mixture of denoising score matching losses of score-based generative models for representation learning. Since the representation encoder is conditioned on time, it can be viewed as an infinite dimensional representation. Results demonstrate improvement over previous semi-supervised learning techniques by pre-training classifiers with the method proposed in this paper.


**Summary Of The Review:**

Authors propose a simple idea to extract multiple-scale features with denoising score matching. However, very similar approaches have already been proposed in previous works and authors fail to cite and discuss.

---

> ### Author Response · Authors · 2021-11-16
> **Authors' response to Reviewer VdM1**
>
> We thank the respected reviewer for the valuable comments. We appreciate the positive feedback that recognizes the contributions of our work. We address the major comments here and take into account the other comments on writing in the revised manuscript.
> RCy: Reviewer’s concern No.y
> AR: Authors’ response
>
> RC1. The idea of representation learning with multi-scale denoising score matching is not new. Very similar ideas have already been discussed in at least [2][3][4].
>
> AR: We thank the reviewer for pointing out the similarity of DRL to the various work using multi-scale denoising autoencoders for representation learning. Due to the similarity in the fact that both Denoising Autoencoders (DAEs) and DRL train representations that can be controlled by the noise schedule, we extended our introduction to also point to the significant works using DAEs. However, we believe there is a significant technical difference between the use of autoencoders in DAEs and the encoder in our method. Denoising Autoencoders extract features directly from the noisy images, while the encoder of DRL extracts features only from the clean images. The representation in DRL depends on the noise level because the decoder is conditioned on the noisy image, which contains partial information about the original image. For example, DRL can be used to only encode fine-grained features when focusing on low noise levels (cf. Figure 2), which is not directly possible with denoising autoencoders. We added a clarification in the Introduction of the revised manuscript to prevent confusion for the readers.
>
>
> RC2. Using "diffusion-based" and "score-based" interchangeably in the paper makes it very confusing. I highly recommend sticking to one name. Given the dependency of the proposed approach on denoising score matching losses, I feel "score-based" might be a more appropriate naming.
>
> AR: We thank the reviewer for pointing out the inconsistency, we now limited the use of “score” to the terms “score matching” and “score function”, and use the term diffusion to refer to the family of diffusion models, which in our perception is more popular than score-based models.
>
>
> RC3. Proposition 1 is quite straightforward. Everyone who understands the proof of denoising score matching should know clearly that it is equivalent to the Fisher divergence up to a constant. It feels inappropriate to write down the expression of the constant as if it were your own contribution.
>
> AR: The reviewer is right that framing it as a proposition is misleading while the aim was to emphasize the non-vanishing constant that is used thoroughly in our work. We changed Section 2.1 in the revised manuscript to put the focus on the importance of the constant constituting the non-zero lower bound.
>
>
> RC4. The second sentence in section 3 is misleading. The work of Song et al. 2021a didn't claim diffusion models are trained to minimize KL-divergence—it is only true for a particular weighting of the score matching losses called the likelihood weighting. Also, the statement holds for the likelihood weighting even if the score model is not curl-free.
>
> AR: We thank the reviewer for emphasizing this important point. We added a clarification that it only holds when using the likelihood weighting. Song et al. 2021a note in the last paragraph of Section 4.2 that the assumption of the model being curl-free is not necessarily true in practice and thus the training only approximately maximizes likelihood. (Song et al. 2021a: “our score model may not be a valid score function [...]. Therefore [...] it is not theoretically guaranteed to make the likelihood of $p^{ODE}_θ$ better.”)
>
>
> RC5. In the second paragraph of section 3.2, the authors claimed that the maximum pairwise distance of images is 170 for CIFAR-10. This is incorrect according to song and ermon 2020, where the maximum pairwise distance is reported to be around 50 for images with pixel values in [0, 1].
>
> AR: We thank the reviewer for pointing out the mistake, the 170 is the maximum pairwise distance after transformations, however, this is not relevant since the noise is added before the transformation and not afterward. We would like to clarify that this error has no effect on our experiments since this number was not used to tune any hyperparameters. We fixed the wrong statement.
>
>
> RC6. The adversarial training of lambda divergences is not principled because the unknown constant of denoising score matching can depend on lambda. Also authors should cite and discuss the related work [1] which discusses adversarial training for score matching losses.
>
> AR: Section 3.1 in the revised manuscript includes a note that the adversary is biased by the constant and thus might not behave as intended. The empirical results in Table 2 however suggest that adversarial training improves the generational performance despite the constant in the DSM objective. We now added a note on previous work on adversarial score matching, since it is certainly related.

---

> ### Author Response · Authors · 2021-11-21
> **Update on concerns**
>
> Please let us know if our responses answered your concerns and convinced you to improve your initial score. We are more than happy to elaborate more on any point that may still be unclear.

---

> > ### Comment · Reviewer_VdM1 · 2021-11-30
> > **Feedback on rebuttal**
> >
> > I would like to thank the authors for their careful response and revision. The rebuttal has addressed some of my initial concerns and I would like to acknowledge this by increasing my review score. However, I still think the novelty of this work is marginal, and the optimization based on lambda-divergence is less principled.

---

### Official Review · Reviewer_XjvM · 2021-11-08

**Correctness:** 3
**Technical Novelty And Significance:** 3
**Empirical Novelty And Significance:** 3
**Recommendation:** 3
**Confidence:** 5

**Main Review:**

The paper is easy to follow and well-motivated. The idea of incorporating a latent code into the score network looks promising and interesting.  However, I have a few concerns:

* A few important formulations and implementation details are missing. For example, what is the exact formulation of VDRL? It is only briefly discussed at the end of section 2.3 but without further introduction. For the semi-supervised learning experiments, what is the value of $t$ used to extract the latent code in inference time? Did you use the adversarial training and initial noise schedule tricks that are introduced in section 3?

*  From the limited experiment description, I suppose that for the semi-supervised learning, the encoder is first pretrained by the DRL objective and then finetuned by the original LaplaceNet algorithm. In that case, it seems that the most contribution leading to the performance is still from the LaplaceNet, and the improvement is tiny. A more informative experiment is to extract features directly from the learned encoder and train an L-2 SVM classifier on the top.

* Section 3 and the corresponding experiment subsections look like a completely different topic that is not relevant to the paper title "representation learning". I would suggest removing those content and including more details and variants of representation learning experiments.



**Summary Of The Paper:**

This paper proposes a revised version of score-based models for representation learning. Specifically, they add an additional input to the score network which is a latent code encoded from the clean sample by an encoder. The encoder can be further conditional on time. Using the pertaining algorithm they achieve improvement of SOTA semi-supervised learning methods. They also show how adversarial training and tuning of the initial noise schedule can improve the performance of score-based models.

**Summary Of The Review:**

Overall I appreciate this paper's idea of using score-based models for representation learning. However, I feel the writing and experiments can be improved before it gets accepted.

---

> ### Author Response · Authors · 2021-11-16
> **Authors' response to Reviewer XjvM**
>
> We thank the respected reviewer for the valuable comments. We appreciate the positive feedback that recognizes the contributions of our work. We address the major comments here and take into account the other comments on writing in the revised manuscript.
> RCy: Reviewer’s concern No.y
> AR: Authors’ response
>
> RC1: A few important formulations and implementation details are missing. For example, what is the exact formulation of VDRL? It is only briefly discussed at the end of section 2.3 but without further introduction. For the semi-supervised learning experiments, what is the value of $t$ used to extract the latent code in inference time? Did you use the adversarial training and initial noise schedule tricks that are introduced in section 3?
>
> AR: We thank the reviewer for pointing out the missing information. We added the formal objective of VDRL and clarified that we did not apply adversarial training and noise scheduling tricks for the semi-supervised learning experiments. The final values of $t$ after training are always in the range of [0.4, 0.6], which is stated in the Appendix A.6.
>
>
> RC2: From the limited experiment description, I suppose that for the semi-supervised learning, the encoder is first pre-trained by the DRL objective and then fine-tuned by the original LaplaceNet algorithm. In that case, it seems that the most contribution leading to the performance is still from the LaplaceNet, and the improvement is tiny. A more informative experiment is to extract features directly from the learned encoder and train an L-2 SVM classifier on the top.
>
> AR: A direct evaluation of the representation using a SVM classifier on the frozen representation for few-shot classification is included in A.5 Figure 12. Please note that unlike contrastive representation learning methods DRL is not specifically designed for the downstream task of classification and is thus not competitive when evaluating the class separation of the representations without finetuning. We added a clarification in the Introduction of the revised manuscript.
>
>
> RC3: Section 3 and the corresponding experiment subsections look like a completely different topic that is not relevant to the paper title "representation learning". I would suggest removing those content and including more details and variants of representation learning experiments.
>
> AR: We appreciate the reviewer’s concern about the coherence of the paper. However, we believe the referred section fits in our work for the reason that follows. Even though our work is mainly focused on representation learning, it is based on diffusion-based training of the score function in the first place. There are important hyper-parameters that we found quite important when in our end-to-end pipeline such as the weighting ($\lambda$ ) and the initial noise scale. The aforementioned remark prepares for the choice of these hyperparameters.

---

> > ### Comment · Reviewer_XjvM · 2021-11-23
> > **Thank you for the response**
> >
> > Thanks for the response from the reviewer(s).
> >
> > The response addresses my first concern.
> >
> > However, for my second concern, after checking the feature learning results in A.6, I think that the learned representations are not satisfactory with low classification accuracy. In fact, I doubt that the improvement upon a pure SVM trained on 100 / 1000 labeled images without pertaining (which is not reported as a natural baseline) is small. The author(s) contrast their work to DAE in the introduction, but no empirical result is shown.

---

> ### Author Response · Authors · 2021-11-21
> **Update on concerns**
>
> Please let us know if our responses answered your concerns and convinced you to improve your initial score. We are more than happy to elaborate more on any point that may still be unclear.

---

### Decision · Program_Chairs · 2022-01-20

**Decision:**

Reject

**Comment:**

The paper proposes extracting multiple-scale features using denoising score matching. Reviewers pointed out the limited novelty in the work and that it does not cite various previous work and how it connects to them.  The paper needs some further  polishing on the writing, and in making the use of lambda divergences more rigorous and principled as explained in the comment of Reviewer VdM1 .